# Does Cannabis Use Contribute to Schizophrenia? A Causation Analysis Based on Epidemiological Evidence

**DOI:** 10.3390/biom15030368

**Published:** 2025-03-04

**Authors:** Sepehr Pourebrahim, Tooba Ahmad, Elisabeth Rottmann, Johannes Schulze, Bertram Scheller

**Affiliations:** 1Clinic of Anesthesiology, Surgical Intensive Care, Emergency Medicine and Pain Medicine, Main-Kinzig-Kliniken, Herzbachweg 14, D-63571 Gelnhausen, Germanylisarottmann0@gmail.com (E.R.); 2Institute of Occupational, Social and Environmental Medicine, Goethe-University Frankfurt/Main, Theodor-Stern-Kai 7, D-60590 Frankfurt/Main, Germany; tooba_91@hotmail.com; 3Paediatric Department, Epsom General Hospital, Epsom and St Helier University Hospitals NHS Trust, Dorking Road, Epsom KT18 7EG, Surrey, UK; 4Anesthesiological Clinic, St. Josef-Hospital Wiesbaden, Beethovenstraße 20, D-65189 Wiesbaden, Germany; bscheller@joho.de

**Keywords:** cannabinoids, psychosis, causality assessment, tetrahydrocannabinol, dihydrocannabidiol

## Abstract

Cannabis abuse has been linked to acute psychotic symptoms as well as to the development of schizophrenia. Although the association has been well described, causation has not yet been investigated. Therefore, we investigated whether cannabis or cannabinoid use is causal for the development of schizophrenia, conducting a systematic literature review according to the PRISM guidelines. Epidemiological studies and randomized clinical trials investigating the links between cannabis and psychosis-like events (PLE) and schizophrenia were identified (according to PRISM guidelines), and relevant studies were included in a Forest plot analysis. Confounder analysis was performed using a funnel plot, and the Hill causality criteria were used to estimate causation. A total of 18 studies fulfilled the search criteria; 10 studies were included in a forest plot. All studies reported an increased risk for PLE or schizophrenia, and nine of the ten studies, a significant increase; the overall OR was calculated to be 2.88 (CI 2.24 to 3.70), with a twofold-higher risk calculated for cannabis use during adolescence. Confounder effects were indicated by a funnel plot. The Hill criteria indicated a high likelihood for the contribution of cannabis to schizophrenia development. Cannabinoids likely contribute to chronic psychotic events and schizophrenia, especially if taken during adolescence. This effect likely increases with a high cannabis THC concentration and increased frequency of cannabis use, and is stronger in males than in females. This points to the possibility of a selective cannabis toxicity on synaptic plasticity in adolescence, as compared to adult cannabis use. Cannabis use should be regulated and discouraged, and prevention efforts should be strengthened, especially with reference to adolescence.

## 1. Introduction

Recreational cannabis use is illegal in most countries, but it is one of the most commonly used drugs, both in frequency and dosage, with estimated 228 million users in 2022 worldwide [1]. Until recently, the main cannabis varieties were dried flowers (marijuana) and resin (haschisch); however, high-tetrahydrocannabinol (THC) varieties like “Sinsemilla” and “Skunk” are marketed in many countries [2]. Besides recreational use, cannabinoids and cannabis flowers are licensed medications in many countries for a variety of diseases [3,4,5]. Despite its status as medication, evidence for its efficiency is scant, and this evidence has not been confirmed in meta-analyses, e.g., findings for spasmolysis in multiple sclerosis [6]. Side effects of cannabis and cannabinoids include acute psychotic states and psychosis-like events (PLE), decreased cognitive functions and attention, and lower concentration [7]. Long-term effects are less well-established; besides drug dependence, neurocognitive deficits, and morphological changes in brain architecture, PLE and schizophrenia have been postulated as direct consequences of cannabis abuse [7,8]. Among these effects, the chronic effects pose a much larger risk for the individual as well as creating a large burden of disease for society.

Schizophrenia is defined by the *Diagnostic and Statistical Manual of Mental Disorders*, Fifth Edition (DSM-5), as a combination of two or more symptoms, including delusions, hallucinations, disordered speech, and disturbed behavior [9]. Substance-use disorder is common among patients with schizophrenia [10,11], and its prevalence is estimated to be about 40 to 50% [12], with even higher values determined for cigarette smoking [12,13,14]. Impaired social functioning is a key criterion for the diagnosis of schizophrenia, and such impairments are highly correlated with deficits in interpersonal skills; additionally, affected individuals have deficits in neuropsychology and social cognition [15], as well as emotional recognition [16]. Khokha et al. [17] describe a high risk of development of substance-use disorders in schizophrenic patients, whereas usually drug abuse—especially cannabis abuse—is observed prior to the diagnosis of schizophrenia [11,18]. For this association a “neurobiological predisposition” has been proposed, as well as a “self-medication” hypothesis [12]. While psychotic symptoms are seen as predominant by some authors [10], others postulate more negative symptoms [12]. Case studies demonstrate a faster development of psychotic symptoms with earlier and/or more frequent cannabis use [19]. Early on, Smit et al. postulated that cannabis use nearly doubles the risk of the development of schizophrenia [20]. The strong correlation between cannabis use and the early onset of psychosis suggests that cannabis use could cause or accelerate the onset of schizophrenia.

No meta-analysis has been published yet for chronic psychotic cannabis reactions. In order to investigate whether cannabis abuse is a major contributor to the development of schizophrenia or merely indicates individuals with an increased risk to develop schizophrenia, we performed a meta-analysis of all relevant studies that calculated an OR for chronic PLE or schizophrenia. Using the Hill criteria for causality we studied whether the association was contributory/causal or merely temporal.

## 2. Materials and Methods

The databases PubMed, Cochrane Library, EMBASE, PsycINFO, Google Scholar, and the American and European Clinical Trial databases https://clinicaltrials.gov and https://www.clinicaltrialsregister.eu (both accessed on 13 July 2024) were searched following the Preferred Reporting Items for Systematic reviews and Meta-analyses (PRISM; this study has not been registered). The search term “(cannabis OR marijuana) AND (schizophrenia OR psychosis) AND (prevalence OR risk OR incidence OR association OR relationship)” was used, with and without the restrictions “RCT”, “observational study”, and “review”. Additionally, citations in reviews and full texts analyzed from original articles were screened for additional studies not identified in the original search. For the further analysis, only English-language publications were included.

This selection resulted in 371 articles, which were then screened for relevance according to titles and abstracts. According to their abstracts, 203 publications did not directly report observational studies or RCT; in 114 articles, schizophrenia or psychosis had not been diagnosed according to ICD or DSM criteria, control groups were lacking, or data sets were incomplete, resulting in the exclusion of 317 articles. The full texts of the remaining 54 articles were downloaded and analyzed by at least two independent reviewers. In sum, 36 additional articles were excluded since the inclusion/exclusion criteria were not met, and/or confounders were not listed. From these eighteen studies, odds ratios from ten studies were included in the forest plot analysis, and eight studies were excluded because of major overlap with another study, a misfit of psychosis criteria or the lack of a control group (Figure 1).

Inclusion criteria comprised the following: -Observational studies, case-control studies, and RCT for the association of cannabis use and the development of psychotic or affective disorders (schizoaffective symptoms);-Psychotic or affective symptoms diagnosed as specified by ICD or DSM criteria;-Anamnestic prolonged cannabis use;-Sufficient description of confounders, e.g., use of other drugs, preexisting disease, or medications.

Exclusion criteria comprised the following:-Case reports or case series;-Overlapping cohorts in multiple publications;-No control group included;-No calculation of odds ratio possible.

### 2.1. Data Extraction and Quality Evaluation

Data extraction was performed by at least two authors for each published article. From all publications, the following information was extracted: authors, publication year, type of study, country of origin, number of controls and patients, method to determine cannabis exposure (if possible, estimation of cumulative dose), diagnostic criterion (ICD, DSM), odds ratios, confidence intervals, and confounders.

To estimate the possibility of bias all included articles were assessed by the criteria of Moore et al. [21] for confounders in randomization (negative if no method is given), frequency of study deviations (number of incomplete data sets), diagnostic criteria (qualitative, semiquantitative), and outcome reporting; additionally, the GRADE criteria [22] were applied for risk of selection bias (e.g., participant selection, participating institutions), inconsistencies (i.e., deviation from comparable parameters, other studies), indirectness (e.g., selection of effects, diagnostic criteria), imprecision (i.e., qualitative reporting), and quality of results (completeness of datasets). A funnel plot analysis was also performed to estimate comparability of the included studies; a larger number of studies outside of the expected funnel indicates a substantial level of study heterogeneity.

### 2.2. Statistical Analysis

Based on 10 articles with a given odds ratio and confidence interval, a forest plot analysis was performed [23]. If no OR value was given in an article it was calculated from the relative risk values for the control and exposed groups, in addition to an estimated confidence interval. To identify additional confounders or other possible bias factors, a funnel plot was constructed [24].

## 3. Results

### 3.1. Study Characteristics

In total, we identified 18 articles investigating the relationship between cannabis use and schizophrenia (Table 1). The oldest article was from 1997, with a nearly identical distribution of publications for the years between 2004 to 2024. Study design, psychosis parameters, and follow-up were heterogenous. Only three studies were longitudinal; they included adolescents 14–16 years of age, and registered psychotic symptoms [25,26] or affective symptoms [27] with 6–10 years follow-up, Callaghan et al. [28] followed adolescents for only 1 year. Additionally, some cross-sectional studies were included in the analysis [29,30,31], as was the study of Power et al. [32] although this twin study focused on genetic factors. Sevy et al. [33] studied risk factors in schizophrenic patients, i.e., reverse causation. In order to estimate an overall risk for schizophrenia by cannabis use we analyzed these studies together despite the large study variability; however, the resulting forest plot estimate has to be discussed cautiously. On the other hand, we excluded studies if no odds ratio could be calculated or where there were variations in study protocols, as well as intervention studies with cannabidiol; the study from Hjorthoj et al. [34] was excluded, since this study analyzed the same cohort as the 2023 study [29]. Studies not included in the forest plot could provide data for the criterion-based causality analysis.

### 3.2. Study Quality

For all studies, the risk of bias was quantified by the criteria published by Moore et al. [21]. Figure 2 illustrates the results of this study; most studies, including seven of the ten studies selected for the forest plot, have some concern for bias, most often in the randomization domain, with the least concern being for missing data.

The ten studies included in the forest plot were assessed for their data quality using the GRADE criteria: “risk of bias”, “inconsistency”, “indirectness”, “imprecision” and “quality” [22]. Table 2 summarizes this analysis, with a low possibility of flaws indicated by a green symbol, a moderate concern identified with a yellow symbol, and insufficient information with a blue symbol.

Most studies were conducted in the US (4 studies), followed by the UK (3 studies), and then Australia, Canada, and Denmark (2 studies each); one study per country was performed in the other countries, or studies were performed in multiple countries. All studies were located in industrialized countries and included high-quality data. Despite their leading roles in cannabis research and decriminalization of cannabis use, and long-existing markets for medical cannabis, no study was performed in Israel [44] or Italy [45].

For the calculation of the forest plot (Figure 3) ten studies were included; since only three studies directly addressed causality of psychosis or affective disorders by cannabis, we combined RCT, case control studies, and cohort studies for this meta-analysis in order to approach the recommended number of studies for a forest plot. The relevant information for these 10 studies is summarized in Table 3.

The forest plot resulted in a determination of an overall OR of 2.88 for the association of psychosis/schizophrenia and cannabis use. By far, the highest risk was calculated by McDonald et al. [25] who restricted the outcome parameter to ER visits and hospitalizations only, i.e., included in this calculation only severely psychotic events. If all psychotic episodes were included, the OR decreased to 11.2 (CI 4.6–27.3); since the study included a cohort of 11,363 adolescents their impact on the overall OR is high; however, calculation with the lower OR value only slightly lowered the overall OR. Patton et al., [27] with a low OR as compared to the other studies, used affective symptoms rather than psychosis; excluding this study would slightly increase the overall risk.

Five studies reported OR values separately for men, and four studies for women, allowing an estimate for sex differences. Not all OR values are comparable. McDonald et al. [25] additionally report an OR for men of 1.22, if women are used as reference. Other studies also found a higher OR for men than for women, with the exception of Patton et al. [27], indicating a higher OR for women for affective symptoms. More consistent are results for the age subgroups, with most studies including adults older than 18 y (legal age of adulthood). Two studies [25,26] included cannabis use in adolescence and found higher OR of 26.7 [25] and 6.5 [26], as compared to studies considering cannabis use later in life. This finding suggests an age-dependent effect of cannabis ingredients on the development of psychosis or schizophrenia.

Since research in cannabis effects may be hampered by a multitude of confounders, we performed a funnel plot analysis to search for unreported confounders. The symmetric distribution of data points in the funnel plot indicated no major misrepresentation; many symmetrical outliers indicate confounders like unequal selection and different end-point stringency, and may also represent the combination of case control studies, prospective studies, and cross-sectional cohort studies in this analysis.

### 3.3. Causality Criteria

Data were examined with the aim of discovering whether they—in addition to a timely correlation—also indicated a causal contribution of cannabis use to later development of psychosis and/or schizophrenia. Hill [46] proposed nine criteria for causality in epidemiological data, with causality being more likely with a rising number of positive criteria. This approach also is amenable for the cohort studies included in this analysis; its results are summarized in Table 4.

Among the nine criteria, five support a causative role of cannabis for psychosis/schizophrenia development; however, experimental proof is lacking since conducting this experiment in humans would be unethical, and no animal model for psychosis exists. Also, plausibility, coherence, and analogy depend on objective clinical criteria in addition to patient reports; the former, however, do not exist for psychotic symptoms. Taken together, the criteria support the causative role of cannabis use/misuse in the development of psychosis.

## 4. Discussion

A connection between cannabis use and later development of psychotic experiences and schizophrenia has been found in a number of studies. Since correlation is no proof of causation, we tried to further investigate this connection with regard to pathophysiological causation.

The most stringent connection is a direct chronic psychotic reaction to cannabis or THC intake, i.e., the exogenous cannabinoid hypothesis [43], as has been well documented for acute cannabis effects [7,35], and even for low doses in human volunteers [35]. Adverse reactions reported in clinical trials do not mention psychosis. Psychotic cannabis reactions are comparable to those resulting from other hallucinogenic compounds, like serotonin [47]; whether these will later develop into schizophrenia—with symptoms occurring without an exogenous trigger—is unclear.

In 2008, Müller-Vahl and Emrich [48] proposed a “cannabinoid hypothesis” of schizophrenia based on the interplay of the activating dopamine receptor system with the modulating endocannabinoid receptor system, which is compatible with cannabis being a relevant factor in schizophrenia development. Accordingly, Murrie et al. [49] found a high rate of transition from acute psychosis to schizophrenia, i.e., from acute substance toxicity to disease induction, with cannabis users showing the highest rate among drug users; however, this study did not address the age of drug consumption or the age of disease onset.

Another hypothesis interpreted cannabis use as an indicator for individuals at high risk of developing schizophrenia; this hypothesis has been coined the “shared vulnerability” hypothesis [50]. Fischer et al. [51] associated altered risk-taking and risk perception in schizophrenic cannabis users; in this interpretation, cannabis users would underestimate risks; also, cannabis use would serve as an indicator rather than a contributing agent. Similarly, a genetic factor for both schizophrenia [52,53,54] and drug addiction [55,56] exists, although in both cases, genome-wide-association studies indicated a polygenetic contribution, with hundreds of genes involved for both schizophrenia and addiction. This makes accidental overlaps likely, and the 3–27 overlapping genes, as identified by Cheng et al. [57], most likely are arbitrary rather than causal.

The most intriguing result in our analysis is the large age effect (adolescent cannabis users) on chronic psychosis/schizophrenia development. Two studies [25,26] recruited adolescents of 14–16 years of age and prospectively studied the occurrence of PLE and schizophrenia over 10 years in relation to cannabis use; both found a high OR, of 26.7 (after adjusting for other risk factors, [25]) and 6.5 [26], whereas cannabis use in adulthood (legal age) had a much lower potency for PLE and schizophrenia. This age sensitivity could be explained by cannabinoid effects on pubertal brain development. Based on the small amount of available data, this phenomenon appears to be specific for PLE, since Patton et al. [27] could not confirm a similar effect for affective disorders like depression or anxiety.

Cannabinoid effects have been found to be dose-dependent. Whereas for acute toxic PLE, this is to be expected, for chronic effects this dose-dependency has not been proven. Contrastingly, Hides ([42]; data not included in analysis due to a very low case number) found the highest psychotic effects at an intermediate dose, and Di Forti et al. ([30]; data included from 2019 publication) found increasing OR with earlier drug intake, high THC content, and frequent cannabis use, all indicating a positive dose dependency.

Therefore, we speculate that cannabinoids have two major psychotic effects: acute psychotic sensations comparable to other hallucinogenic drugs and indicative of acute toxicity; and altered synaptic plasticity during adolescence.

The current evidence also supports a higher vulnerability in women, as compared to men [25,29].

Therefore, we propose the specific contribution of cannabinoids, especially THC, to the development of schizophrenia. The effect is likely larger when THC exposure occurs in adolescence, compared to prenatal exposure [58] or adult exposure (this analysis); it may be caused by changes in synaptic plasticity, and it possibly affects specific patient subgroups.

### Limitations

Our hypothesis is based on the available epidemiological evidence; the contribution is strengthened by the coherence of data across different study designs and outcome criteria, as well as by a strong support for causality by the Hill criteria. The proposed hypothesis is in accordance with data for brain development, and the comparable effects from other hallucinogens. However, human evidence is limited both in the number of studies as well as their comparability; only two studies directly address causality in a prospective study design. Also, diagnostic criteria vary across studies, with some studies including patients with self-described psychosis-like experiences, while others only consider hospital treatment of schizophrenia. These differences result in a large heterogeneity. Both basic research and epidemiological studies are necessary, especially to confirm adolescence as a specifically critical exposure time, and to determine the impacts of THC content and duration of abuse as criteria for dose dependency.

## 5. Conclusions

Cannabinoids likely carry a specific contributing risk for the development of schizophrenia and PLE in later life, especially when consumed during adolescence. Preventing cannabis use in this life stage may have a large effect on reducing the burden of psychiatric diseases. Drug prevention programs for adolescents should be strengthened, and the availability of cannabis strictly regulated.

## Figures and Tables

**Figure 1 biomolecules-15-00368-f001:**
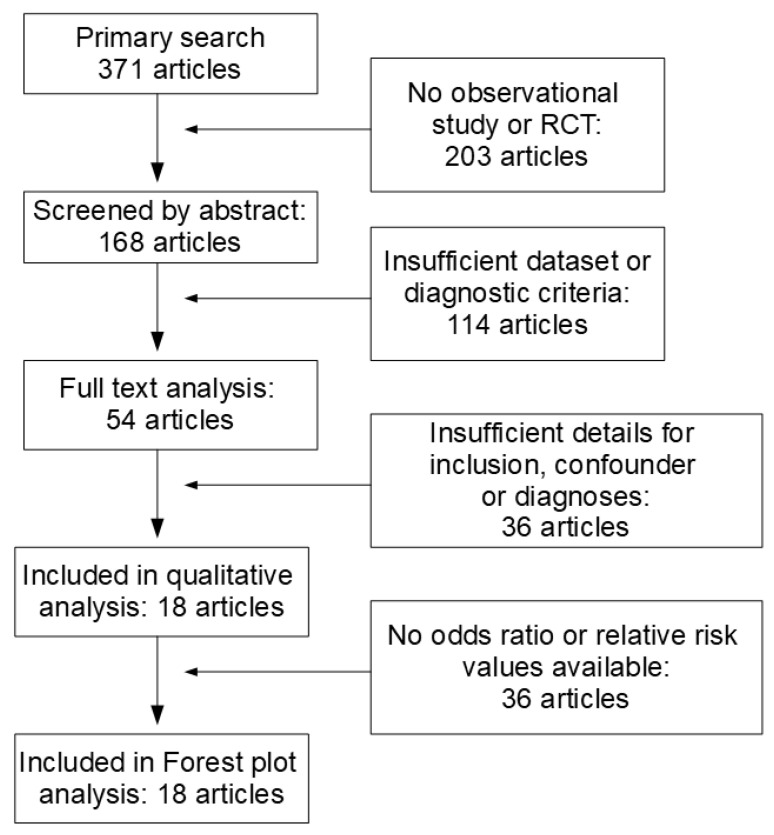
PRISM flow chart for study inclusion and exclusion factors.

**Figure 2 biomolecules-15-00368-f002:**
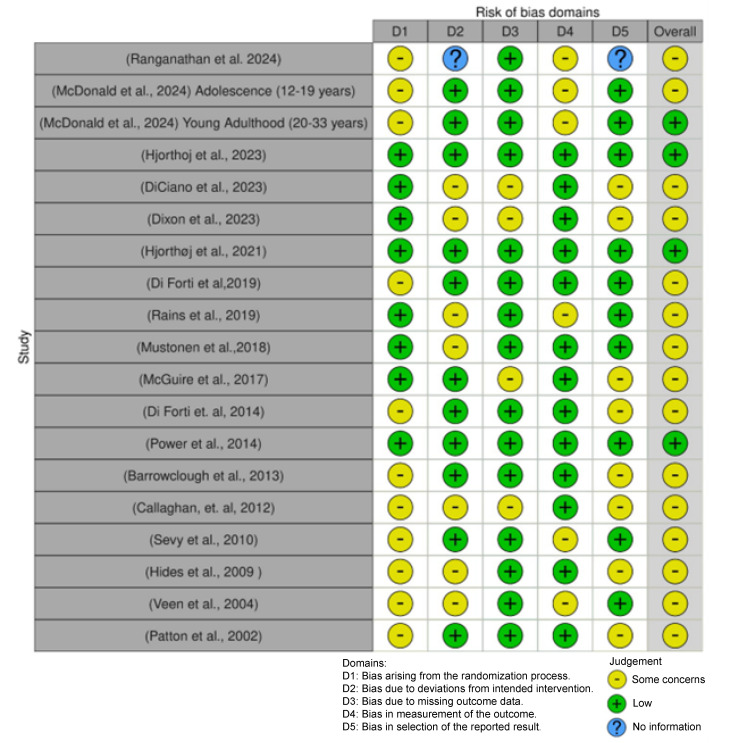
Possibility of bias in the publications identified in the literature search [25,26,27,28,29,30,31,32,33,34,36,37,38,39,40,41,42,43].

**Figure 3 biomolecules-15-00368-f003:**
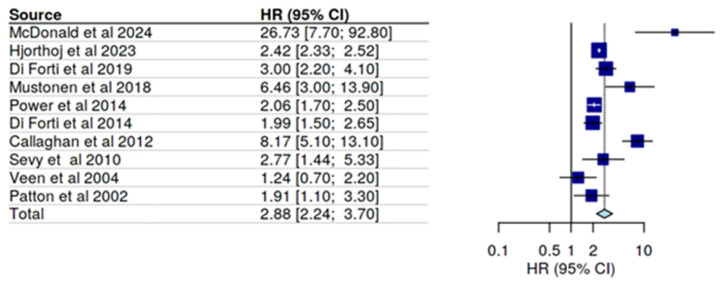
Forest plot for the association of psychosis/schizophrenia and cannabis use [25,26,27,28,29,30,31,32,33,38].

**Table 1 biomolecules-15-00368-t001:** Articles investigating the correlation between cannabis use and psychosis/schizophrenia.

Reason for Inclusion/Exclusion	Focus of Study	Country, Participants	Type of Study	Study
Excluded; no odds ratio given	Acute psychotic events after 10 mg THC	U.S.A., 31 participants	RCT	Aghaei et al., 2024 [35]
Included	Development of psychosis after cannabis use in adolescence	Canada, 11,363 part., 12 to 24 y	Longitudinal cohort study	McDonald et al., 2024 [25]
Excluded; psychosis was no end-point	Impairment of driving after smoking cannabis	Canada, 31 older volunteers	Open label exposure study	Di Ciano et al., 2024 [36]
Excluded due to study protocol, no results yet	Decrease in psychosis by CDB	U.S.A., 120 patients	RCT	Dixon et al., 2023 [37]
Included	Correlation of schizophrenia diagnosis and cannabis use disorder	Denmark, 7,186,834 population, 16–49 y	Population study	Hjorthoj et al., 2023 [29]
Excluded; cohort identical to that of Hjorthoj et al., 2023 [29]	Population-attributable risk factor of cannabis for schizophrenia	Denmark, 6,907,859 population, 16–49 y	Population study	Hjorthoi et al., 2021 [34]
Included	Correlation of psychosis and cannabis use in psychiatric patients	EU, Brazil; 901 cases, 1237 control	Case control study	Di Forti et al., 2019 [38]
Excluded; no odds ratio for psychosis given	Effectiveness of cannabis-based medicine in cannabis reduction	London UK, 278 interv., 273 controls	RCT	Johnson et al., 2019 [39]
Included	Development of psychosis after cannabis use in adolescence	Northern Finland, 6534 patients 16 y	Longitudinal population study	Mustonen et al., 2018 [26]
Excluded; no THC applied	Antipsychotic activity of 1000 mg CDB	Europe, 43 interv., 45 controls	RCT	McGuire et al., 2017 [40]
Included; small cohort overlap with Di Forti et al., 2019 [38]	Association of psychosis and cannabis use in psychiatric patients	London, UK, 410 cases	Cohort study	Di Forti et al., 2014 [30]
Included; similar/identical genetics	Association of schizophrenia with cannabis use in twins	Australia, 2082 participants	Twin registry	Power et al., 2014 [32]
Excluded; no control group without drug abuse	Comparison of different drug-abuse personalities	UK, 160 users, 167 nonusers	Case control study	Barrowclough et al., 2013 [41]
Included; “Cannabis-related problems” over 1 year	Correlation of cannabis use with CRP	BC, Canada; 3339 users	Population study	Callaghan et al., 2012 [28]
Included	Risk factors and hallucinations in cannabis users	USA, 49 cases, 51 controls	Case control study	Sevy et al., 2010 [33]
Excluded; very low case number	Psychosis in drug abuse, 19 regular cannabis users	Australia; 881 adolescents 16 y	Cohort study	Hides et al., 2009 [42]
Included	Frequency of cannabis use in schizophrenic patients, 15–54 y	The Hague, The Netherlands	Cohort study	Veen et al., 2004 [31]
Included	Development of depression/anxiety in cannabis users	Australia, 1601 students 14–15 y	Longitudinal cohort study	Patton et al., 2002 [27]

**Table 2 biomolecules-15-00368-t002:** Study quality of studies included in the forest plot.

Quality	Imprecision	Indirectness	Inconsistency	Risk of Bias	Study
Serious	Serious	Serious	Not serious	Not serious	McDonald et al., 2024 [25]
Moderate	Not serious	Not serious	Serious	Not serious	Hjorthoj et al., 2023 [29]
Not serious	Not serious	Serious	Serious	Serious	Di Forti et al., 2019 [38]
Not serious	Serious	Serious	Serious	Serious	Mustonen et al., 2018 [26]
Not serious	Not serious	Serious	Serious	Serious	Di Forti et al., 2014 [30]
Moderate	Serious	Not serious	Not serious	Not serious	Power et al., 2014 [32]
Moderate	Serious	Not serious	Not serious	Serious	Callaghan et al., 2012 [28]
Not serious	Serious	Serious	Not serious	Serious	Sevy et al., 2010 [33]
Moderate	Serious	Not serious	Not serious	Serious	Veen et al., 2004 [31]
Not serious	Serious	Serious	Serious	Serious	Patton et al., 2002 [27]

**Table 3 biomolecules-15-00368-t003:** Overview of the human studies for cannabis use and psychosis/schizophrenia included in the forest plot (Figure 3).

Confounders	OR (CI)	Parameter	Criteria/Intervention	Country, Participants	Type of Study, Psychosis Criteria	Study
Sex (m > f), income, race, urban, cigarettes, alcohol, illicit drugs	12–19 y: 26.7 (7.7–92.8) >19 y: 1.8 (0.6–5.4)	Physician or ED visit for psychotic disorder	Questionnaire, yes/no last year, Kaplan–Meier analysis	Canada, 11,363 part., 12 to 24 y	Longitudinal cohort study, DSM/ICD	McDonald et al., 2024 [25]
Alcohol, drug abuse, psychiatric history	Male: 3.84 (3.43–4.29) female 1.81 (1.53–2.15)	Psychiatric hospitalization	Register study, PARF, incidence rates	Denmark, 7,186,834 population, 16–49 y	Population study, ICD	Hjorthoj et al., 2023 [29]
Prior psychosis, SES	3.2 (2.2–4.1)	First episode of psychosis	Cannabis use, PARF	EU, Brazil; 901 cases, 1237 control	Case control, ICD-10	Di Forti et al., 2019 [38]
Alcohol or other drugs, family history, SES	6.5 (3.0–13.9)	Psychosis	Cannabis use	Northern Finland, 6534 patients, 16 y	Prospective population study, ICD	Mustonen et al., 2018 [26]
Age, ethnicity, SES, illicit drugs	1.39 (1.11–1.68)	Psychosis	Cannabis use,	London, UK, 410 cases	Cohort study, ICD-10	Di Forti 2014 [30]
SES, gender	2 (1.7–2.5)	Schizophrenia	GWAS, logistic regression	Australia, 2082 part.	Twin registry, ICD	Power et al., 2014 [32]
Sex, age, education, income, SES	Male > female, young, income, marital state	Cannabis-use disorder	Cannabis use, logistic regression	BC, Canada, 3339 users	Population study, DSM	Callaghan et al., 2012 [28]
Gender, age, SES, education	2.77 (1.44–5.33)	Schizophrenia	CUD/non-CUD	USA, 49 cases, 51 controls	Case control, DSM	Sevy et al., 2010 [33]
Age at milestones	2.5 (1.1–5.0) males/female	Psychosis incidence	Drug use	The Hague, The Netherlands, 70 cases	Cohort study, ICD	Veen et al., 2004 [31]
Alcohol, tobacco, illicit drugs	Daily: ♂ 1.9 (0.93–3.8) ♀ 8.6 (4.2–18)	Depression, anxiety	Cannabis use, frequency	Australia, 1601 stu dents 14–15 y	Longitudinal cohort study, DSM	Patton et al., 2002 [27]

**Table 4 biomolecules-15-00368-t004:** Application of Hill criteria to epidemiological support for cannabis causality towards schizophrenia.

Main Reasons	Causality Support	Criterion
All studies positive	Strong	Strength of association
All studies positive	High	Consistency
Different psychosis criteria between studies	Moderate	Specificity
Three longitudinal studies	High	Temporality
Dose-dependent effects if dose is quantified	High	Biological gradient—dose response
No pathophysiological model for psychosis or schizophrenia	Moderate	Plausibility
No positive criterion for psychosis	Moderate	Coherence
Unethical studies	Lacking	Experiment
Similarity with dopamine-induced psychosis	Moderate	Analogy

## Data Availability

No new data were created or analyzed in this study. Data sharing is not applicable to this article.

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
