# Peer review of "Does Cannabis Use Contribute to Schizophrenia? A Causation Analysis Based on Epidemiological Evidence"

_biomolecules, 2025, doi:10.3390/biom15030368_

Round 1

Reviewer 1 Report

Comments and Suggestions for Authors

The study examined the causal relationship between cannabis abuse and the onset of schizophrenia, or whether it merely signifies individuals predisposed to schizophrenia, while identifying genetic or social risk factors. However, there are few comments which need to be addressed below:

Abstract:

Overly detailed for an abstract, particularly in reporting specific OR values (e.g., "OR of ~6 to 7").

The causality criteria explanation is too technical for an abstract.

The abstract lacks implications for future research or practice, which could enhance its impact.

Simplify the results section to highlight key findings without technical details and add implications or recommendations.

Introduction:

The introduction is overly long and includes some redundant points (e.g., description of cannabis types and legality could be condensed).

Lacks a clear statement of how this study fills a gap in existing literature.

Need to focus more on the research gap and rationale for the study, reducing general information.

Methods:

The study lacks details about the screening process (e.g., whether two reviewers independently screened studies).

Risk of bias assessment is mentioned but not described in sufficient detail.

Require to include more details about the selection process, risk of bias tools, and inter-rater reliability if applicable.

Results:

The results section is overly focused on numerical data without enough interpretation.

Study heterogeneity is mentioned but not adequately explored.

Discussion:

The discussion is repetitive, especially about adolescence as a critical period.

Some points (e.g., comparison with other hallucinogens) seem tangential.

Conclusion:

The conclusion does not provide actionable recommendations for policymakers or clinicians.

Include practical recommendations for public health strategies or areas for further research.

Comments on the Quality of English Language

The quality of English is acceptable in the manuscript

Author Response

Reviewer 1:

Comment:

Overly detailed for an abstract, particularly in reporting specific OR values (e.g., "OR of ~6 to 7").

Response: We have deleted details from the methods and results, including the confidence interval.

Comment: The causality criteria explanation is too technical for an abstract.

Response: We simplified the causality criteria by only referring to their common descriptor.

Comment: The abstract lacks implications for future research or practice, which could enhance its impact.

Response: We have added a sentence for further research need.

Comment: Simplify the results section to highlight key findings without technical details and add implications or recommendations.

Response:  We hope to have sufficiently shortened and simplified the Abstract.

Introduction:

Comment: The introduction is overly long and includes some redundant points (e.g., description of cannabis types and legality could be condensed).

Response: We have shortened the introduction, e.g. by eliminating (illegal) cannabis types. However, we consider the varying legality among countries and the double use as medication and recreational drug important.

Comment: Lacks a clear statement of how this study fills a gap in existing literature.

Response: We included a statement that no review has been published for chronic psychosis by cannabis use; we also modified the last paragraph stating the research question of a risk estimate, and the contribution versus association controversy.

Comment: Need to focus more on the research gap and rationale for the study, reducing general information.

Response: We hope that the mentioned changes are sufficient.

Methods:

Comment: The study lacks details about the screening process (e.g., whether two reviewers independently screened studies).

Response: We have specified these details, including two author literature assessment.

Comment: Risk of bias assessment is mentioned but not described in sufficient detail.

Response: We have included the bias criteria both from GRADE and Moore et al.

Comment: Require to include more details about the selection process, risk of bias tools, and inter-rater reliability if applicable.

Response: Inter-rater variability is not relevant, selection criteria for exclusion of studies are specifically given in Table 1.

Results:

Comment: The results section is overly focused on numerical data without enough interpretation.

Response: The research question requires a numerical estimate for the odds ratio; in order for the reader to properly judge the resulting OR we consider the Tables and Forrest plot necessary. We have put the interpretation into the discussion section.

Comment: Study heterogeneity is mentioned but not adequately explored.

Response: We have added a paragraph explaining the results from a Funnel plot analysis, indicating no hint for selective publishing but many outlying studies indicating a variety of bias factors. We hope that this additions will be sufficient.

Discussion:

Comment: The discussion is repetitive, especially about adolescence as a critical period.

Response: We have shortened the specific section; however, since the large difference betweenm using cannabis in adolescence versus adulthood in our opinion justifies the emphasis on this point. We hope to have eliminated the redundancies.

Comment: Some points (e.g., comparison with other hallucinogens) seem tangential.

Response: We have shortened the comparison section, and have removed the paragraph dealing with molecular actions of THC and CBD.

Conclusion:

Comment: The conclusion does not provide actionable recommendations for policymakers or clinicians.

Include practical recommendations for public health strategies or areas for further research.

Response: We have added the strong recommendation of cannabis use prevention and health information; however, since at least in Europe cannabis consumption is fraught with political impact additional general recommendations e.g. for banning cannabis for all countries seems to be impossible.

Reviewer 2 Report

Comments and Suggestions for Authors

This is a very interesting and a timely paper and analyses, which investigated whether cannabis and cannabinoid use is "causal" for the development of schizophrenia. However, given the extremely complex neurodevelopmental and pathophysiological aspects of schizophrenia, with the most multi-etiological causes (of any diseases affecting mankind), it is almost impossible to determine any specific cause for schizophrenia.

However, for sure, cannabis use is a significant factor to "contribute" to the causation of schizophrenia. The last paragraph/conclusion of this article exactly stated the same which is an excellent final paragraph:

The paper is well presented, however my suggestions are as below:

-Please consider to revise the title; is state that ; " Does cannabis use "contribute" to cause schizophrenia?

-In the text, where applicable, please stress and note that cannabis is contributory rather than sole cause of schizophrenia (other wise as noted in this paper we would have additional over 228 million patients with schizophrenia world wide, if all users develop schizophrenia).

Author Response

Reviewer 2:

Comment: However, for sure, cannabis use is a significant factor to "contribute" to the causation of schizophrenia. The last paragraph/conclusion of this article exactly stated the same which is an excellent final paragraph:

Response: Throughout the manuscript we have changed "causal" to "contributory"

The paper is well presented, however my suggestions are as below:

Comment: Please consider to revise the title; is state that ; " Does cannabis use "contribute" to cause schizophrenia?

Response: We have changed the title accordingly.

Comment: In the text, where applicable, please stress and note that cannabis is contributory rather than sole cause of schizophrenia (other wise as noted in this paper we would have additional over 228 million patients with schizophrenia world wide, if all users develop schizophrenia).

Response: See above.

Round 2

Reviewer 1 Report

Comments and Suggestions for Authors

Comments have been addressed meticulously.

Reviewer 2 Report

Comments and Suggestions for Authors

Revisions are satisfactory.  Thank you.